# Used like Pawns or Treated like Kings? How Narratives around Racehorse Welfare in the 2023 Grand National May Affect Public Acceptance: An Informed Commentary

**DOI:** 10.3390/ani13193137

**Published:** 2023-10-08

**Authors:** Gemma Pearson, Janet Douglas, Inga Wolframm, Tamzin Furtado

**Affiliations:** 1The Horse Trust, Slad Lane, Princes Risborough, Buckinghamshire HP27 0PP, UK; 2Easter Bush Campus, University of Edinburgh, Midlothian EH25 9RG, UK; 3World Horse Welfare, Anne Colvin House, Snetterton, Norwich NR16 2LR, UK; janetdouglas@worldhorsewelfare.org; 4Applied Research Centre, Van Hall Larenstein University of Applied Sciences, Larensteinselaan 26-A, 6882 CT Velp, The Netherlands; inga.wolframm@hvhl.nl; 5Leahurst Campus, University of Liverpool, Neston, Liverpool CH64 7TE, UK; tfurtado@liverpool.ac.uk

**Keywords:** Grand National, racing, equine, welfare, social licence, horse sport, human behaviour change

## Abstract

**Simple Summary:**

The Grand National is one of the world’s most famous steeplechase races. In 2023, the start of the race was delayed, as an animal rights group protested about the race by gaining access to the course. When the race eventually took place, several horses fell, with one sustaining a fatal injury. The ensuing commentary from racing afficionados and animal activists alike laid bare some of the fundamental issues surrounding the sport. This event highlights how the racing industry is increasingly subject to public pressure around the safety and welfare of horses. Public acceptance of activities such as racing is known as “social licence to operate” (SLO) and, as societies and cultures evolve, so too does the concept of social licence; for example, the SLO around circus animals has largely been “lost”, while zoos have managed to maintain their SLO through shifting their focus towards education and the conservation of wild animals. Following the 2023 Grand National, pro- and anti-racing groups shared their views on a variety of media platforms. In this commentary, we consider how the narratives presented might shape the future of the racing industry.

**Abstract:**

The 2023 Grand National steeplechase race was delayed when protesters from the animal rights group, ‘Animal Rising’, gained access to the course just prior to the race. The international media spotlight was focused on what is already a high-profile event and the social licence of both this race and racing in general was scrutinised. Both at the time and for several days afterwards, the general public was exposed to two different narratives from pro- and anti-racing communities. This paper discusses these perspectives and the potential impact on the general public’s relationship with racing. Whilst well-meaning and aiming to promote racing, much of the racing industry’s commentary inadvertently risked damaging its reputation due to a poor understanding of social licence principles. We explore the reasons for these two groups’ alternative perspectives on welfare and suggest considerations for change. Ultimately, if ‘the people’s race’ is to maintain its social licence, the racing community needs to both understand and embrace the concept. Welcoming independent opinions, engaging with different viewpoints, accepting that change is inevitable and, most importantly, being proactive in making changes to prioritise equine welfare will all help racing to move towards greater public acceptance.

## 1. Introduction

The Grand National, a steeplechase held over a course of 4 miles and 514 yards (6.9 km) and 30 jumps at Aintree Racecourse every April, is arguably the most iconic race in the British horseracing calendar. Dubbed ‘the people’s race’ and ‘the most famous horse race in the world’, television audiences frequently exceed 7.5 million in the United Kingdom alone [1]. Much of the Grand National’s attraction can be attributed to the combination of its large and unusual jumps [2] and big field of around 40 competitors. This results in a spectacular test of skill, stamina, and bravery of both human [3,4,5,6] and equine [7,8] athletes, with an element of luck thrown in for good measure. However, what might pass as entertainment for some is cause for concern to others. With an average of 0.75 equine deaths/year since 1950 (Figure 1), the high level of risk to its equine competitors has turned the Grand National into a contentious sporting event.

Over the years, there has been growing public disquiet over the risks of equine injury and death in jump racing in the United Kingdom [10], and in the Grand National in particular [11,12]. The visibility of this race means that it attracts particular attention, and it is probably for this reason that the activist group, Animal Rising, staged a protest prior to the 2023 Grand National, with several protestors gaining access to the track and attempting to attach themselves to fences. One of Animal Rising’s aims was to prevent the race from taking place, thereby preventing any equine fatalities [13]. Their protest resulted in a 15 min delay to the start of the race, during which the horses were first returned to the pre-parade ring, with some being unsaddled only to be tacked up again moments later. The disruption caused by the protesters, the death of a horse during the race, and the number of horses that fell, have led to extensive discussion and media coverage—from the racing community and animal rights protesters alike—around the future of horse racing. 

This paper discusses the television commentary that was aired at the time of the race, as well as points raised subsequently in mainstream and social media. Using theory and research relating to social licence, equine welfare, and human psychology, it provides an informed commentary on the power of public opinion to control the future of an industry and the advisability of taking the views of society into consideration in public discourse. It also promotes some ideas that racing may find helpful as it moves forward in this era of increased public sensitivity to animal welfare.

## 2. Social Licence to Operate, Human Behaviour, and Animal-Related Sports

It is evident that public perception regarding the use of animals for human entertainment is changing [14]. This has led to the reduction, modification/regulation, or complete cessation of a range of animal-use activities (Table 1).

These changes, which reflect both increases in scientific knowledge surrounding animal welfare and evolution of ethical belief systems [24], were all associated with a growing sense of unease with the status quo and/or an incident or incidents that galvanised public sentiment [25,26,27,28,29,30,31,32]. In this context, if not handled well, the events surrounding the 2023 Grand National could herald substantial changes to jump racing’s regulatory environment. 

The link between the regulatory environment of an activity and its public acceptance is mediated to a large extent by its social licence to operate (SLO) [33]. Originally coined in relation to the mining industry, the term and associated concept of ‘social licence’ is now applied extensively to animal-use activities, including sport [33]. A social licence is a dynamic, intangible, unwritten contract between society and those who are involved in an industry or activity [34,35]. It describes the degree of public acceptance of the industry/activity, and society’s trust in its proponents to behave ethically, legally, and responsibly [33,35]. While the concept of social licence may appear to be somewhat abstract, with no formal legal standing, it can nevertheless be extremely powerful in altering the future of an activity.

Loss of social licence typically starts with negative portrayal of the activity/industry in the media [33]. If the proponents of the activity do not take effective action, this may lead to loss of political support, revised legislation, and—potentially—an outright ban [33] (Figure 2). It is important to note that once an industry’s social licence has been substantially damaged or lost, it can be difficult to regain [36]. Jump racing in the United Kingdom is currently faced with negative portrayal in the media. However, in some jurisdictions, horse racing is already faced with revised legislation [20] and in others jump racing has been banned completely [37]. Proponents of racing in the United Kingdom should therefore regard the current situation as a warning of potential future restrictions to their sport.

If an animal-use sport is facing a challenge to its social licence—as is the case for jump racing in the United Kingdom in 2023—animal welfare (or the perception thereof) frequently lies at the heart of the problem [33,38,39]. This is certainly the case for horse racing, as well as equestrianism more generally, as shown by a number of surveys and informed commentaries [26,40,41]. A television and radio debate following the 2023 Grand National suggested that there is minimal public support for the race to continue in its current form [42]. Those expressing their distaste for the race included many who had previously supported it, with welfare concerns forming the basis of their change in sentiment [42]. The proportion of negative texts sent to the show led the debate’s host to suggest that this reflects changes to societal norms and attitudes and to state that ‘the tide is turning’ against racing. 

The SLO concept is often considered in isolation (e.g., [33]). However, in reality, maintenance of social licence requires an interdisciplinary approach involving multiple aspects of scientific theory followed by practical application of the findings. If jump racing is to improve its social licence, placing equine wellbeing at the heart of its activities, these disciplines should include equine welfare science, as well as principles of human psychology and behaviour change. Welfare science allows us to understand how the affective (emotional) states that horses may experience as a consequence of their management and interactions with people affect their quality of life [43]. The scientific fields of human psychology, social cognition, and human behaviour change, on the other hand, allow us to understand why humans think and act the way they do and to put in place strategies that encourage and enable people to modify their behaviour [44,45,46]. In this paper, we explore these diverse yet interlinked fields as they relate to the events of the 2023 Grand National.

## 3. Communicating with the Public about the Grand National

When listening to the publicly available narrative on the day of and after the Grand National, one of the most striking observations was the difference between the main messages conveyed by racing commentators and Animal Rising representatives. Animal Rising had made it clear that their intention was to stop the race to prevent a catastrophic injury from occurring [47]. However, rather than immediately addressing this discussion point during the delay prior to the start of the race, the television commentary team focused primarily on the message that racehorses have the best welfare possible when in training/away from the racetrack [48]. The commentators subsequently encouraged anyone with concerns about the welfare of racehorses to visit a racing yard and see for themselves how well the horses are cared for. Yet the fact that changes had been made to the Grand National course and fences to reduce the risk of injury to the horses was only briefly referred to 15 min into the commentary. After a further 5 min, there was more airtime given to this topic, but it was still minimal. 

Two days after the race, when mainstream and social media in the United Kingdom were filled with concern about the risks faced by racehorses when jump racing [49,50], representatives from Animal Rising and racing came together in a national radio show [42]. As Animal Rising continued to emphasise the risks to which the horses were exposed during a race, racing’s representatives were once again reluctant to address Animal Rising’s main point. Although one racing stakeholder did mention the work undertaken on risk reduction, the discourse from the racing representatives was dominated by the message about racehorses’ quality of life away from the racecourse, as well as the implications for the welfare of horses and people working in racing if racing were to be stopped. Various members of the general public who were involved in the debate commented that they were concerned about the risk to the horses in the Grand National, regardless of how well they were being looked after at home, and questioned why the racing community only talked about racehorse welfare at home and were failing to discuss safety improvements. In this debate, the majority of comments made by the general public (by phone-in or text message) related to concern about the risk of injury to the horses during racing. In order to maintain racing’s social licence, this topic should be publicly acknowledged by the sport’s proponents. 

Animal Rising’s stance could arguably be taken as a proxy—albeit an extreme one—for growing public concern about the fate of racehorses during racing. If the racing industry wishes to continue to enjoy public support, it is vital that it not only takes these concerns seriously, but that it is seen to be taking them seriously. While the sport’s leaders have undoubtedly taken steps towards increased safety, including making changes to the Grand National which were initially associated with a substantial reduction in the number of horse falls over these fences [8], the main communication emphasis remains on the horses’ quality of life outside of the actual races.

In terms of the racing industry’s achievements on equine safety, overall, fatalities in racing (flat and jump) in Britain have reduced by a third over the past 20 years to a current figure of 0.2% of runners [51]. Research is now moving at pace to analyse data relating to the ‘Jump Race Risk Model’, a project that aims to determine risk factors for injury in jump racing Thoroughbreds [52]. For the Grand National specifically, there has been extensive analysis of data around risk factors for falls, some fences have been lowered and/or modified, the distance to the first jump has been reduced, the ‘going’ is carefully monitored and modified if necessary, the jockeys are briefed regarding speed to the first fence, the minimum age for horses to compete has been increased, and there have been changes to the horse and jockey entry criteria [53,54]. 

However, these efforts notwithstanding, the faller and fatality rates in jump racing remain stubbornly high, with latest figures showing rates of 2.25% (1 in 44) and 0.43% (1 in 233), respectively [55]. In terms of social licence, publicly acknowledging the nature and seriousness of problems, being humble and transparent about the issues, and engaging with critics as well as supporters on how to improve equine safety and welfare can lead to benefits that may come as a surprise to industries that have not traditionally acted in this way [56]. This contrasts with much of the narrative surrounding the 2023 Grand National, in which the ‘cross purposes’ arguments may only have turned off the general public and led to the conclusion that racing is trying to divert attention away from the incidence of catastrophic injuries. This approach will not move the conversation forward, nor will a relentless focus on ‘positive messaging’ help to maintain racing’s social licence—a lesson that has been learned, to their cost, by numerous other industries [57,58].

After all, the way human beings interpret the world around them is largely shaped by the information regarding that world to which they have access. The more easily certain viewpoints, incidents, or opinions can be recalled, the more likely it is that those viewpoints, incidents, etc. are considered an accurate reflection of reality [59,60]. This so-called ‘availability bias’ [59] is influenced, and even shaped, by (social) media [61,62]. The more media attention a certain topic attracts, the more readily it will be recalled by the general public. At its most extreme, this can lead to an ‘availability cascade’, a self-sustaining chain of events kicked off by the reporting of an event which causes an emotional reaction within the general public. The reaction of the public becomes newsworthy itself, escalating to the point that it becomes a matter of public policy to interfere [63]. The impact of this phenomenon on equestrian sports has recently been highlighted [64].

In the current climate, equine fatalities that occur during high profile races are highly emotive and result in extensive media coverage. Consequently, the perception of those individuals with little to no previous experience of racing will likely be shaped by those media reports as that is the information most readily available to them. This makes it all the more important, therefore, that the racing industry recognises the real risk associated with negative media coverage.

## 4. ‘Walking the Walk’: Prioritising Welfare

Much of the current commentary around racing seems to suggest that racehorses experience the best possible welfare. Phrases such as ‘they are looked after like kings’, ‘they stay in 6* hotels’, and ‘they receive 5* treatment from the day they are born until the day they die’ are not uncommon and were widely used during the television coverage of the 2023 Grand National [48]. A British Horseracing Authority statement subsequent to the race even described ‘providing an unparalleled quality of life for horses bred for racing’ [65]. However, given current scientific understanding of horses’ ethological and physiological needs, how credible is this messaging? 

Welfare scientists are clear that if good welfare is to be achieved, access to the ‘3 Fs’ (friends, forage, and freedom) should form the basis of a horse’s life [66]. This involves having the opportunity to interact with and touch/groom other horses, having access to forage for the majority of the day, and being able to ‘be a horse’ (i.e., to graze, socialise, and move at liberty). Whilst many aspects of racehorse welfare continue to improve (e.g., it appears to be increasingly popular for racehorses to have access to turnout), many animals still have restricted access to these basic needs [67]. As a consequence, the prevalences of abnormal repetitive (stereotypic) behaviours (a coping mechanism for stress) and gastric ulcers within the racehorse population are high [67,68]. For example, the prevalence of squamous gastric ulcers is typically considerably higher in racehorses than in other equine populations [68]. These behavioural and health problems are an indicator of poor welfare.

If the racing industry is to maintain credibility and show that it is ‘walking its talk’ when it comes to providing excellent equine welfare, it is vital that common management practices are re-evaluated in the light of current scientific knowledge. What is more, where there is room for improvement, this should be publicly acknowledged and followed by widespread change across the industry. Welfare should be evaluated using a horse-centred approach, with a willingness to accept advice from independent welfare scientists. This would demonstrate that the industry takes its responsibility to its horses seriously, is competent to look after their best interests (both mental and physical), and is transparent about its practices—all factors that will help to maintain public trust and, thereby, racing’s social licence [33]. 

One notable example of good practice came from Lucinda Russell, trainer of the 2023 Grand National winner, Corach Rambler. Russell declined the opportunity to parade the horse at other race fixtures after his win, saying that he was now on his ‘holidays’. Instead, she posted pictures on social media of him in the field with another horse, thereby providing him with the 3Fs. Approaches such as this, which rely on “showing” rather than “telling”, are likely to engage others in more meaningful dialogue and knowledge transfer [69], practices, and communications. In addition, the public are able to distinguish between genuine attempts to optimise welfare based on current scientific understanding versus unsubstantiated positive messaging [38]. Therefore, actions such as Russell’s will go much further in terms of maintenance of social licence than verbally asserting that racehorses enjoy the highest standards of welfare, whilst many of them continue to be denied their basic ethological needs.

## 5. Embracing Shared Values

Much of the public criticism of racing in general—and the Grand National in particular—has centred on concerns about equine injuries and fatalities [13,42,49], culminating in the current demands to minimise risk in such events or to ban them altogether. It is important to note, however, that the fundamental value that underpins such sentiments is the protection of the horses’ welfare. One of the most important tenets of social licence is demonstrating shared values with stakeholders [70]. Since an industry’s stakeholders include those individuals or groups who can affect the industry [35], Animal Rising and other critics of the sport are relevant stakeholders. Understanding, engaging with, and then moving towards these values is therefore an important step in maintaining racing’s SLO. The effectiveness of this approach is illustrated by work conducted in the food industry, which showed that adopting shared values is 3–5 times more important in building public trust (and therefore maintaining social licence) than demonstrating competence [71]. 

Moreover, although racing’s proponents and opponents may appear to be fundamentally at odds with each other, the industry and many anti-racing groups actually share an important value: reducing the incidence of equine injuries and fatalities. However, whilst this overall value is shared, there are clear differences in the level of risk that is deemed acceptable. For example, during the pre-race commentary, it was mentioned that changes have been made to make the sport safer, suggesting that those within the industry want to see fewer injuries [48]. But after highlighting these changes, racing’s commentators said: ‘welfare and safety is the best it has ever been’, in a tone that implied that the current level of risk is acceptable. An alternative and more appropriate approach might have been to emphasise that there is still much more work to be done. The current level of risk was further downplayed by stating that ‘all sport is dangerous, racing is no different’, trivialised by comments such as ‘they can lose their life in a field, half a ton of animal on the skinniest of legs’ and misrepresented by a statement that horses are at no greater risk when racing in the Grand National than when they are turned out [48]. Given that, typically, one in 40 horses sustains a catastrophic injury over the course of this 10 min race, this would suggest that if a trainer cares for 40 horses, all of which have access to turn out, one would expect one horse to be seriously injured for every 10 min of turn out time—the equivalent of 6 fatal injuries per hour. This is obviously not true, so statements such as this do nothing to promote public confidence that racing’s proponents understand the values of the sport’s critics or are serious about tackling the issue of equine injury and fatality. 

It is of course unrealistic to expect that the sport of jump racing will ever achieve an injury rate of 0%. But that provides no justification for not focusing relentlessly on moving in the direction of that goal. Not only is this a result that racing’s proponents themselves would like, but it would show clear adoption of a fundamental value of many of the sport’s opponents. A relentless focus on safety has underpinned impressive reductions in risk in both Formula 1 racing [72] and the energy sector [73] and could help to defuse criticism and maintain public acceptance of racing. Certainly, there is ample evidence that opponents of an industry, including some advocacy non-governmental organisations (NGOs), will work constructively with the proponents of the industry if they are seen to be making genuine attempts to improve the situation [74,75]. Engaging with moderate critics, acknowledging difficulties, and being transparent about progress is often an effective means of bridging the gap between an industry and its opponents [56].

## 6. Understanding Other People’s Perspectives

We have discussed racing’s focus on informing the public that racehorses experience excellent welfare, illustrated by comments such as “racehorses live like kings”, and assertions that their every need is attended to by staff who love them. There is little question that the majority of the people who work within the racing industry do indeed love horses and this was portrayed well by the ITV media team. However, many of racing’s critics also love horses and—also like racing’s proponents—they are motivated by the desire to provide horses with good welfare. Conflict arises because of a difference between the two sides in underlying ethical belief systems, with groups such as Animal Rising aiming to stop the 2023 Grand National from running because of the risk of equine injury and fatality [13], and racing’s supporters accepting the current level of risk as being acceptable on the basis that they believe the horses are well looked after otherwise [48].

The gap between the two viewpoints seems vast, and the extreme stance of some anti-racing NGOs appears to be a barrier to meaningful discussion with many of racing’s proponents. However, industries can glean useful information from such organisations about the issues that concern them, since their views undoubtedly reflect the opinions of at least a subset of the public [56] and are therefore relevant to the industry’s future. In addition, seeking to work with more moderate NGOs and critics of an industry—and advertising that fact—can be an effective way of ‘reducing space’ for those with more extreme viewpoints [56]. Discussing with such groups the shared value of safeguarding equine welfare, while also openly acknowledging differences in underlying belief systems, would facilitate discussion which could ultimately bring together the opposing communities and open channels to improve equine welfare. Even more enlightened is to invite advocacy groups to become involved in setting standards within an industry and auditing policies, practices, and impacts [56].

## 7. Communicating and Engaging with Critics

Maintenance of SLO relies on communities with different views being prepared to acknowledge—if not understand—each other’s perspectives and to discuss their respective positions in a constructive way. It is also important that both sides are willing to engage in compromise and possible change [76]. If this is to occur, the barriers to effective communication must first be identified and overcome. Studies show that people with strong views on a particular subject often align their sense of identity with their beliefs [77,78]. As a result, a “threat” to the belief in the form of disagreement or an opposing argument may be perceived as threatening a person’s identity, causing resistance to change. In addition, studies have shown that providing information that opposes strongly held beliefs of an individual or community may actually strengthen those beliefs, as well as leading to frustration with those who provided the information [79,80]. Therefore, in the face of strong moral views, providing information (i.e., ‘educating’ critics) is often not the most effective strategy for bringing about changes in attitude or behaviour. Instead, theorists posit an approach known as “moral reframing”, in which each group considers the values which are relevant to the opposing group and reframes their arguments in the context of those values [81]. This demonstrates a willingness to engage in open discussion and compromise, rather than simply arguing for one view, and can lead to establishment of two-way communication and meaningful discussion [58]. 

Following the 2023 Grand National, both the racing industry and anti-racing protesters used arguments which they felt were compelling in order to assert their position. However, it was clear in the narratives presented that there was divergence between the two groups in terms of their perspectives. As an example, Animal Rising believe that horses should not be used for the purpose of human entertainment under any circumstances. In contrast, the majority of racing’s proponents are happy for horses to race for entertainment providing they have a good quality of life [48,82,83]. In this situation, the arguments posited by both groups show a strong sense of identity as caring and loving horse people. The ensuing debates involving stakeholders who held these two opposing views led to little meaningful discussion, with each group’s identity threatened by the other. As a consequence, some racing commentators discredited the identity of the protesters by suggesting that they would not be able to handle racehorses successfully and could not be “horse lovers” [48]. As a result of this approach, both sides reasserted and re-presented their own beliefs in subsequent media debates, rather than addressing one another’s concerns [42]. 

Given the studies exploring changes in attitude mentioned above, it is clear that this approach is unlikely to be successful in altering others’ perceptions or attitudes and may simply strengthen each group’s original beliefs and sense of identity. Moreover, for members of the general public who are beginning to form doubts over their support of the Grand National, exposure to strong arguments based on loose facts may push them further away from racing, despite the commentators’ aim of doing the opposite. Open discussion which is respectful of the identity, knowledge, and beliefs of both sides could lead to more meaningful discussion and collaboration. One example of a more open approach was shown by the trainer Peter Scudamore, who suggested: “We [racing] should not say we need to educate them [the protestors] as it makes it sound like we are better than they are” [48], a statement that acknowledges the need to communicate from a platform of mutual respect. Another positive example would be the ITV media team repeatedly acknowledging that the protesters have a right to a peaceful protest, and, subsequently, racecourses offering them a designated area from which to protest that would not impact on horse welfare [48,84]. 

## 8. Attacking the Attacker vs. Inviting the Wolf into the Henhouse

A common theme running through racing’s commentary on the Grand National was to belittle the protestors and their views [48]. On the day of the race, protestors were dismissed as being “attention seekers” who do not love horses and who did not have a serious point to make, and it was also suggested that protesters “need educating” about racehorse welfare [48]. Two days after the race, a representative from Animal Rising was repeatedly asked for his plans on how he would manage the 50,000 Thoroughbreds currently in the United Kingdom if racing were to cease [42]. Whilst this is undoubtedly a valid discussion point, most experts would contend that it is not the job of an industry’s critics to find solutions to issues—that is a job for the industry itself, with the majority of NGOs seeing their role as raising concerns on behalf of those who have no voice [56].

Many of racing’s proponents have likely been a part of the industry from a relatively young age and derive much of their life’s purpose from it. As has been demonstrated in different studies and across various industries, the development of purpose and personal identity often go hand in hand [85,86]. Whenever individual identity is perceived to threatened, however, this will drive an individual to attempt to diminish or even eliminate the (perceived or real) identity threat [87]. This behaviour (in essence, discrediting one’s opponents or “attacking the attacker”) is a commonly used but ineffective and ultimately unhelpful strategy, both when it comes to dealing with a (perceived or real) threat to one’s identity and when dealing with those who seek to change an industry [71,74]. What is substantially more effective [74,75] is to engage with opponents, learn about their values and ambitions, and—potentially somewhat controversially—even invite them to become part of the solution. This latter technique was used successfully by the fast-food chain, McDonald’s, at a time when they were coming under fire for their waste management and use of polystyrene. McDonald’s invited the Environmental Defense Fund (an environmental NGO) to partner with them on improving their environmental sustainability. The result was a win for both the environment and McDonald’s [75,88]. McDonald’s also had the foresight to engage with some of the most prominent animal rights campaigners in the world, again with positive results for the company [75,88]. Finding itself with substantial challenges to its SLO, McDonald’s sought to learn from and work with its opponents, not to discredit them. 

Dismissal of one’s opponents’ views shows an unwillingness to engage and a lack of respect for those with different viewpoints. This is an unhelpful strategy, particularly given that Animal Rising’s views on the acceptability of the current fatality rate in the Grand National are not inconsistent with those of the substantial proportions of both the general public and those within equestrianism who are currently questioning the welfare of horses in sport [40,41]. Trying to undermine Animal Rising’s credibility and dismissing their views therefore shows no attempt to share the values of a broad swathe of the public, nor a willingness to engage in behaviour change which might protect equine wellbeing, and is not consistent with maintenance of SLO. 

## 9. Embracing Transparency 

A common response of industries to challenge is to ‘pull up the drawbridge’, shunning the voices of critics and treating the vast majority of the industry’s operational information and data as confidential. This is almost certainly a mistake [33,56,89]. As stated by Arnot in 2016, transparency is ‘no longer optional’ if an industry wishes to maintain its social licence [71]. In this context, it is notable that radical and brave approaches to transparency have been highly effective in terms of improving SLO for a number of industries. Examples include Australian Eggs, which has enjoyed a steady increase in positive consumer sentiment in association with adoption of a transparent approach to its operations and challenges [90], and McDonald’s, who tackled challenges to their welfare, sustainability, and environmental credentials by opening their operations up to independent scientists and NGOs [75,88]. This approach was highly effective in restoring public trust and has been described as “[starving] their critics of oxygen and … a demonstration of how to maintain social licence” [74].

## 10. Avoiding “Whataboutery”

It is tempting, in the face of criticism, to point the finger at other branches within one’s own industry, implying that the situation in that branch is worse than in one’s own and that critics should redirect their ire [91]. An example of such ‘whataboutery’ occurred during the television commentary prior to the Grand National, with the statement “It’s a shame they [protestors/critics] do not stand by the roadside/field by ill-kept horses rather than these” [48]. This communication strategy does little to deflect criticism from its original target and may simply result in the public hearing “We might be bad, but they are worse”. Other industries that involve disparate branches have found that working together and taking collective responsibility for issues raised by the public has been an effective strategy in improving social licence [92,93].

## 11. Working with Advocacy NGOs

A social licence is ‘granted’ by the public and it is ultimately their level of trust that dictates the future of an industry. Public perception is not synonymous with that of advocacy NGOs—but those NGOs would not exist and be in a position to mount a challenge if they did not have the support of at least a sector of the public [56]. It is, therefore, in an industry’s best interests to make a serious attempt to engage with them in an open-minded, humble, curious, and non-judgmental way [94]. Most NGOs recognise that their demands are unlikely to be met in full and will work with an industry that acknowledges difficulties and is serious and transparent about its attempts to institute change [56]. Organisations that work in the corporate sector, where social licence is a well-studied issue, have identified a number of common themes that run across industries. These are summarised in Table 2 as a guide to some of the strategies that equestrianism’s regulators and decision makers may consider as they navigate the road ahead.

## 12. Take-Home Suggestions

Based on information from other industries, we suggest that the racing community should consider how both their actions and their way of communicating may shape the future of horse sport. Embracing and engaging with the concerns of the general public and committing to maximising the wellbeing and safety of horses involved in racing are key. Figure 3 summarises the most relevant information presented in previous sections of this paper and suggests ways that the racing industry could maximise public acceptance of the sport as it moves forward in the UK’s evolving culture around animal participation in sport. 

## 13. Conclusions

By gaining access to the course and delaying the start of the 2023 Grand National, the Animal Rising protestors shone a light on this race. National Hunt (jump) racing is already walking a tightrope to maintain its social licence and much of the on-the-day commentary and post-race discussions from racing stakeholders on mainstream and social media may have hindered rather than helped this cause. As animals are increasingly recognised as sentient beings and public perceptions of how we interact with and use animals change, the goal posts to maintain SLO will also change—and will continue to do so—for all industries. This was evident from the overall tone of public opinion aired during the Nicky Campbell ‘Five Live’ debate [42]. This included comments from stakeholders who had previously enjoyed the race who were now voicing concerns, and who suggested that the Grand National is losing its SLO. 

It is natural to feel threatened by change. Given that working in racing often becomes more a way of life than a job, it is likely that, for many people working in the industry, racing’s status quo becomes synonymous with their personal identity. Consequently, there may be a temptation to rely on a combination of discrediting anyone that challenges one’s current beliefs and promoting a positive image of the sport. Sadly, this strategy almost inevitably backfires as SLO is based on trust, not the mere perception of a particular image. The public will not easily have the wool pulled over their eyes, and they can tell the difference between genuine attempts to improve welfare and empty positive messaging. Instead, being brave enough to welcome independent opinion, engage with those with a different viewpoint, accept that change is inevitable in any aspect of life and, most importantly, be proactive in making changes to prioritise equine welfare, using a horse-centred approach from birth to death, will gain public trust and help to maintain a healthy SLO.

## Figures and Tables

**Figure 1 animals-13-03137-f001:**
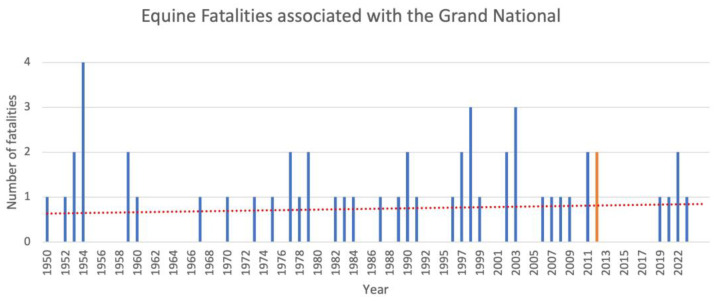
Equine fatalities per year that occurred during the running of the Grand National between 1950 and 2023. Years in which the race did not run are not included. The orange line represents equine fatalities that occurred in 2012, after which changes were made to the race with the aim of improving equine safety. The dotted red line represents the line of best fit (the line that most accurately describes the change over time in the number of fatalities per year). Please note that this graph is intended to reflect what the general public may perceive and therefore shows the absolute number of horses that were named as fatalities each year [9] and does not account for other confounding variables of risk such as number of starters (a factor that would also determine the fatality rate).

**Figure 2 animals-13-03137-f002:**
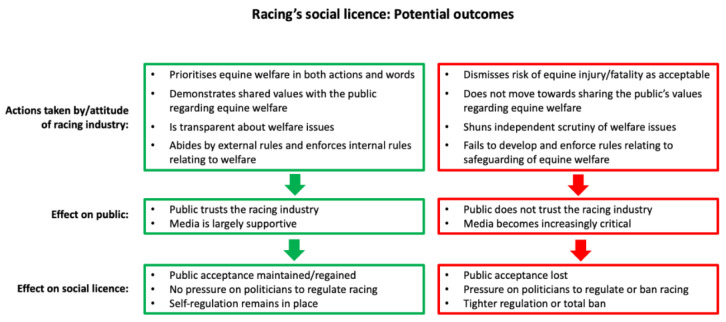
The figure uses two potential scenarios to show how the actions of the racing industry might impact on the sport’s social licence.

**Figure 3 animals-13-03137-f003:**
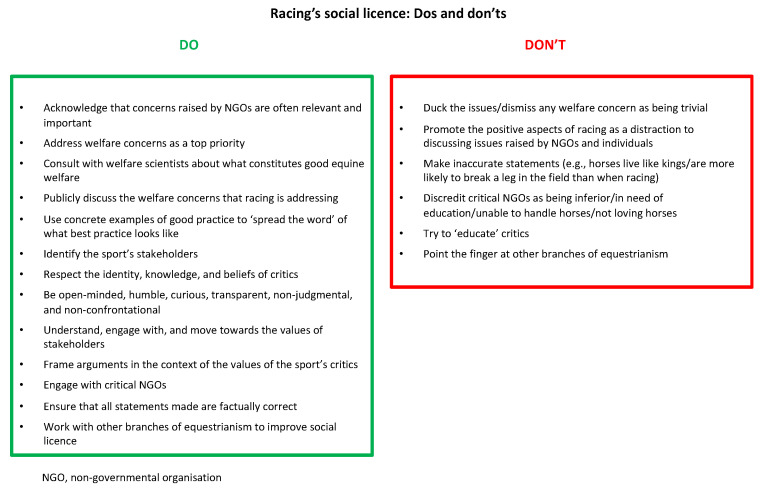
Suggestions for racing industry based on information from other sectors.

**Table 1 animals-13-03137-t001:** Examples of changes to a variety of industries that were driven by social licence.

Reduction	Modification/Regulation	Complete Loss
Betting on horseracing and attendance at racetracks in various countries [15,16,17]	Fox hunting in the United Kingdom [18]	Equestrianism in the Olympic Modern Pentathlon [19]
	Horse racing in the United States [20]	Greyhound racing in the Australian Capital Territory [21]
	Keeping of cetaceans in captivity [22]	The use of wild animals in circuses in the United Kingdom [23]

**Table 2 animals-13-03137-t002:** Working with advocacy NGOs: Strategies that help to maintain social licence in the corporate world [56].

Strategies That Help to Maintain Social Licence in the Corporate World
Recognise that the concerns raised by NGOs are often relevant and important
Appreciate that engaging with credible NGOs: ○ Helps to keep the industry aligned with society’s expectations ○ May allow identification of issues before they become a serious problem
Seek to work with NGOs that are willing to be part of the solution: ○ Focus on shared aims/areas of mutual concern and use these to build a constructive partnership ○ Establish agreement on any disputed facts as an early goal ○ Focus on the process, not the outcome
Involve NGOs in setting industry standards and auditing industry operations
Invite NGO members to sit on industry boards and take their ideas seriously
Involve NGOs in setting benchmarks for improvement
Be transparent about the way in which difficult issues are handled
Adopt a transparent, balanced, open-minded, humble, engaged, and non-confrontational approach in all encounters and communications

NGO, non-governmental organisation.

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
