# Peer review of "Used like Pawns or Treated like Kings? How Narratives around Racehorse Welfare in the 2023 Grand National May Affect Public Acceptance: An Informed Commentary"

_animals, 2023, doi:10.3390/ani13193137_

Round 1
Reviewer 1 Report
This is an excellent and timely paper. Equestrian sport (both for showing & for racing - of all types) is at a crossroads. Examining social license to operate principles is of paramount importance.
There are very minor edits/suggestions/questions on the attached file.

A few very minor punctuation notes were highlighted on the attached file.
This is an excellent and timely paper. Equestrian sport (both for showing & for racing - of all types) is at a crossroads. Examining social license to operate principles is of paramount importance.
There are very minor edits/suggestions/questions on the attached file.
Author Response
Thank you for your rapid and thorough review, and for your positive and constructive comments. Our responses to your specific comments are listed below.
Lines 47/48 – Highlighted text with comment ‘Though I potentially agree, this is a more opinion-based statement’
“While we feel that a statement such as this is acceptable in an opinion paper, and likely to keep the readers interested, we have included references to support our statement”
Line 61 – punctuation here. On the version we submitted there was one full stop. The version you received had already undergone editing from the journal. This is now resolved on the latest copy, thank you for bringing this to our attention.
Line 73/74 - not certain why there is no line of spacing in between these paragraphs versus others.
Line spacing has already been edited by the journal to their style. From what I can see line spacing only occurs at the start of a new section or after a figure/image.
Line 218 – check punctuation. Thank you for bringing this to our attention. The extra full stop was not present on the submitted version and is now corrected.
Line 237 – at least suboptimal. Thank you but we believe that if we used the term ‘suboptimal’ it would suggest that these behaviour and health problems are acceptable. We believe it is not acceptable that they occur with such high prevalence and have therefore retained the word ‘poor’.
Line 352-354 ‘In contrast, the majority of racing’s proponents are happy for horses to race for entertainment providing they have a good quality of life’ - May need to provide a reference; if one for horses can't be found, there is a rather recent one related to dairy cattle and 'a good life' - I think C Dwyer was one of the authors. Relevant refs added
Line 481 ’easily have the wool pulled over their eyes’ not sure if this expression is globally understood <?> Thank you, we have checked with people internationally (including an 8yr old Dutch child!) and everyone has understood the term. However we have added more text to ensure it is understood.
Reviewer 2 Report
Line 61 - remove one punctuation mark (as two exist)
Line 71 - "This," - means what? Rephrase sentence please
Line 75 -Social license change to social licensing ?
Line 78 - "this" - is what? Be specific and clear please
Line 89 - instead of lost perhaps ceased or stopped?
Line 94 - it has changed in Australia with international racehorses requiring scintigraphy before racing to reduce injuries, fatalities and falls in the Melbourne Cup (if that is of interest to the authors).
Line 101 - ethically responsible ?
Lines 154-169 - any media references ?
Line 184 , change to ;
Line 189 - the fall and fatality rates
South Australia and New South Wales in Australia passed legislation to ban jumps racing in Australia due to the risks.
Line 239 - "Walking the talk" - depends on the data you refer to OR your ethical stance. As you show, one trainer has posted the 3F's with the winning horse. There are some trainers that do already recognise the biopsychosocial lens of training horses in racing and the 3 F's. Are there many? Possibly not, if you examine the steward reports and bans from racing for behaviour and banned substances used. However, "should" the trainers listen to welfare scientists? How do we entice them to work collaboratively? Do we approach the Racing licensing industries, the trainers, track riders, marketing, (the entire lot)? E.g. the one welfare lens (horse, rider/owners/trainers/environments)?
LIne 303 - risk mitigation - shorten the course, situational ethical calls, modify the jumps ect per data analysis and weather/track conditions. Despite this, they still had deaths at WEG in France 2014 on the eventing course.
The site of a catastrophic injury and euthanasia on any course or shown on media distresses much of the public/animal lovers.
The conclusion is introducing new information. Please ensure it is overtly stated in the body of the text. Concluding remarks are usually a summary.
General comments
The manuscript discussed the dichotomous experiences between pro racing industry people and the animal activists. It's likely the public in the middle of the normal distribution curve will be the people who hold the numbers to decide what is the social norm. I found this part absent in the manuscript.
Agreed involving stakeholders of all areas is necessary. Transparency is a must.
In Australia, we also have a lifetime registration that follows the horse's life until death when they are re-homed by Racing Victoria accredited re-trainers. There are also special awards for "Off the track series" in showing and dressage. The "special" awards are quite popular for TB's post racing.
Overall, interesting manuscript that is worth publishing with some minor changes. Of note, we need to approach the racing industry with the degree of being humble, as the trainers may need with the activists to effect sustainable change.
Well done!
Author Response
Thank you for your rapid and thorough review, and for your positive and constructive comments. Our responses to your specific comments are listed below.
Line 61 - remove one punctuation mark (as two exist) On the version we submitted there was one full stop. The version you received had already undergone editing from the journal. This is now resolved on the latest copy, thank you for bringing this to our attention.
Line 71 - "This," - means what? Rephrase sentence Now rephrased for clarity
Line 75 -Social license change to social licensing ? Social licensing would be an incorrect term. However we have re-phrased the sentence to hopefully improve readability.
Line 78 – "this" - is what? Be specific and clear please Updated
Line 89 - instead of lost perhaps ceased or stopped? Changed for clarity
Line 94 - it has changed in Australia with international racehorses requiring scintigraphy before racing to reduce injuries, fatalities and falls in the Melbourne Cup (if that is of interest to the authors).
Thank you for your comment, that is very helpful and we are aware of this; after discussion, we prefer to leave the text as is here so as not to detract from the point in this paragraph, (especially as it is controversial - many orthopaedic surgeons don’t believe it reduces the risk but is being used to make it look like they are ‘taking preventative measures’ but take your point! Thank you.
Line 101 - ethically responsible ? Altered to include ethically
Lines 154-169 (174)- any media references ? We have added references to mainstream media each time a comment about something that was said/printed is made. We are not aware of any current method of referencing social media posts, which we did not specifically discuss anyway.
Line 184 , ‘We are unable to see where on line 184 the insertion of a semi-colon would be appropriate – no change made to text’.
Line 189 - the fall and fatality rates. Thank you for the suggestion but faller would be the correct terminology as used in the industry, for example see BHA website on risk
South Australia and New South Wales in Australia passed legislation to ban jumps racing in Australia due to the risks. Yes, the general consensus amongst racehorse vets (and other stakeholders) in the industry is that jumps racing has not got long to last in the UK.
Line 239 - "Walking the talk" - depends on the data you refer to OR your ethical stance. As you show, one trainer has posted the 3F's with the winning horse. There are some trainers that do already recognise the biopsychosocial lens of training horses in racing and the 3 F's. Are there many? Possibly not, if you examine the steward reports and bans from racing for behaviour and banned substances used. However, "should" the trainers listen to welfare scientists? How do we entice them to work collaboratively? Do we approach the Racing licensing industries, the trainers, track riders, marketing, (the entire lot)? E.g. the one welfare lens (horse, rider/owners/trainers/environments)?
Thank you for sharing your thoughts here. These types of questions and considerations are what our current paper tries to elicit, as we believe that by examining and engaging with the different ethical, moral and behavioural currently in evidence in racing and across equestrianism, we can move towards a new set of shared values. We hope that this paper will start a conversation around exactly what you describe, and bring about some industry discussion about those ethical stances you rightly mention. Thank you for bringing this up
Line 303 - risk mitigation - shorten the course, situational ethical calls, modify the jumps ect per data analysis and weather/track conditions. Despite this, they still had deaths at WEG in France 2014 on the eventing course.
The site of a catastrophic injury and euthanasia on any course or shown on media distresses much of the public/animal lovers.
We agree, however we have specifically avoided suggesting risk mitigation as that is not the focus of this paper. Furthermore, as different factors have varying levels of evidence supporting them it becomes controversial – and we want all stakeholders to read this and consider what we say, not be put off.
The conclusion is introducing new information. Please ensure it is overtly stated in the body of the text. Concluding remarks are usually a summary.
Thank you, we have updated the ms and believe it is improved due to this feedback.
General comments
The manuscript discussed the dichotomous experiences between pro racing industry people and the animal activists. It's likely the public in the middle of the normal distribution curve will be the people who hold the numbers to decide what is the social norm. I found this part absent in the manuscript.
Thank you for your comment. We have discussed this at great length, and experimented with different options. We prefer to leave as is; this is because it is likely a combination of the middle ground and the extremist views, given that the middle ground are often the ones most affected by extremist views. Therefore, adding in further information here creates a layer of complexity which we don’t feel is appropriate and we don’t believe we can back up with any evidence.
Agreed involving stakeholders of all areas is necessary. Transparency is a must.
In Australia, we also have a lifetime registration that follows the horse's life until death when they are re-homed by Racing Victoria accredited re-trainers. There are also special awards for "Off the track series" in showing and dressage. The "special" awards are quite popular for TB's post racing.
Thank you for your comment, that is good to hear. It is interesting to compare Australia and the UK. Retraining of Racehorses is immensely popular in the UK now. This had led to them being almost too fashionable and not always ending up with people who have the required experience for them.
Overall, interesting manuscript that is worth publishing with some minor changes. Of note, we need to approach the racing industry with the degree of being humble, as the trainers may need with the activists to effect sustainable change. We strongly agree with this point, thank you
Reviewer 3 Report
A very good commentary.
It is a bit long. There are many repetitions. I think your suggestions could have a bigger impact if you tightened up the text
Author Response
Thank you for your rapid review, and for your kind comment.
A very good commentary.
It is a bit long. There are many repetitions. I think your suggestions could have a bigger impact if you tightened up the text
Many thanks for your comment. We have discussed this as a team, and explored areas for removal. However, following this discussion and the comments of the other three peer reviewers, we prefer to leave as is.
Reviewer 4 Report
Granted this is a commentary rather than a scientific report, a little more detail on what was actually reviewed to portray the media and public responses would be of interest.
On line 86, "horse racing in the United States" does not indicate what change has been transpired. The same point of confusion applies to some of the other examples that are not followed by parentheses describing the change. Either that should be added or better yet, put the examples into a table.
Figure 2 is too small, making it difficult to read. It could be readily enlarged.
Line 237 states that the higher prevalence of gastric ulcers as well as stereotypic behavior are indicators of poor welfare. The word, "poor" seems harsh. Perhaps "suboptimal" would be an acceptable alternative.
Figure 3 was duplicated in the version that was presented to the reviewer.
Author Response
Thank you for your rapid and thorough review, and for your positive and constructive comments. Our responses to your specific comments are listed below.
Granted this is a commentary rather than a scientific report, a little more detail on what was actually reviewed to portray the media and public responses would be of interest.
Many thanks for your comment. While we take your point, this was meant to be a commentary/opinion piece and not a scientific report, and thus we prefer to leave as is so as not to add unnecessary complexity. A future more rigorous systematic review or thematic analysis would be fantastic, but here we are simply exploring concepts.
On line 86, “horse racing in the United States” does not indicate what change has been transpired. The same point of confusion applies to some of the other examples that are not followed by parentheses describing the change. Either that should be added or better yet, put the examples into a table.
Thank you, we have now added a table and hope this provides sufficient clarification
Figure 2 is too small, making it difficult to read. It could be readily enlarged.
Thank you, again this is ultimately the remit of the journal editors but we will raise this with them.
Line 237 states that the higher prevalence of gastric ulcers as well as stereotypic behavior are indicators of poor welfare. The word, "poor" seems harsh. Perhaps "suboptimal" would be an acceptable alternative. We respectfully disagree. Suboptimal suggests it is acceptable. We believe that these measures represent a failure to meet a horse’s basic needs, resulting in what is mainly avoidable pathology and so is consistent with ‘poor’ welfare.
Figure 3 was duplicated in the version that was presented to the reviewer.
Thank you, there was only 1 figure 3 on the submitted version but post editing from the journal there were 2. We have now removed the extra one.